# Abnormal Liver Function Tests and Long-Term Outcomes in Patients Discharged after Acute Heart Failure

**DOI:** 10.3390/jcm10081730

**Published:** 2021-04-16

**Authors:** Hiroshi Miyama, Yasuyuki Shiraishi, Shun Kohsaka, Ayumi Goda, Yosuke Nishihata, Yuji Nagatomo, Makoto Takei, Keiichi Fukuda, Takashi Kohno, Tsutomu Yoshikawa

**Affiliations:** 1Department of Cardiology, Keio University School of Medicine, Tokyo 160-8582, Japan; m.hiroshi.1018@gmail.com (H.M.); white_cascade_libra@yahoo.co.jp (Y.S.); kfukuda@a2.keio.jp (K.F.); 2Division of Cardiovascular Medicine, Kyorin University School of Medicine, Tokyo 181-8611, Japan; ayumix34@yahoo.co.jp (A.G.); kohno-ta@ks.kyorin-u.ac.jp (T.K.); 3Department of Cardiology, St. Luke’s International Hospital, Tokyo 104-8560, Japan; hatasuke@luke.ac.jp; 4Department of Cardiology, National Defense Medical College, Tokorozawa 359-8513, Japan; y.nagatomo1111@gmail.com; 5Department of Cardiology, Saiseikai Central Hospital, Tokyo 108-0073, Japan; makoto_tk@hotmail.com; 6Department of Cardiology, Sakakibara Heart Institute, Tokyo 183-0003, Japan; tyoshi@shi.heart.or.jp

**Keywords:** acute heart failure, liver dysfunction, liver function test

## Abstract

Abnormal liver function tests (LFTs) are known to be associated with impaired clinical outcomes in heart failure (HF) patients. However, this implication varies with each single LFT panel. We aim to evaluate the long-term outcomes of acute HF (AHF) patients by assessing multiple LFT panels in combination. From a prospective multicenter registry in Japan, 1158 AHF patients who were successfully discharged were analyzed (mean age, 73.9 ± 13.5 years; men, 58%). LFTs (i.e., total bilirubin, aspartate aminotransferase or alanine aminotransferase, and alkaline phosphatase) at discharge were assessed; borderline and abnormal LFTs were defined as 1 and ≥2 parameter values above the normal range, respectively. The primary endpoint was composite of all-cause death or HF readmission. At the time of discharge, 28.7% and 8.6% of patients showed borderline and abnormal LFTs, respectively. There were 196 (16.9%) deaths and 298 (25.7%) HF readmissions during a median 12.4-month follow-up period. The abnormal LFTs group had a significantly higher risk of experiencing the composite outcome (adjusted hazard ratio: 1.51, 95% confidence interval: 1.08–2.12, *p* = 0.017), whereas the borderline LFTs group was not associated with higher risk of adverse events when referenced to the normal LFTs group. Among AHF patients, the combined elevation of ≥2 LFT panels at discharge was associated with long-term adverse outcomes.

## 1. Introduction

Liver dysfunction mediated by hemodynamic instability (cardiohepatic interaction) is known to be present in 20–40% of patients with acute heart failure (AHF) [1,2,3]. Analyses from large-scale randomized controlled trials have demonstrated that abnormal values of liver function tests (LFTs) are associated with increased morbidity and mortality in patients with heart failure (HF) [4,5]. However, previous studies have focused largely on a single abnormality of LFTs, such as total bilirubin (TB), aspartate aminotransferase (AST) or alanine aminotransferase (ALT), and alkaline phosphatase (ALP), which may be elevated due to factors other than hemodynamic instability (e.g., drug-induced liver injury, physiological jaundice, and malnutrition). As such, the results were highly heterogeneous, and the clinical relevance may have been overestimated in patients with HF. Overall, the assessment of LFTs in combination, rather than any single parameter, may be more useful for clear decision-making with regard to risk stratification and the tailoring of treatment for HF. To date, there has been no consensus on the type of LFT derangements that is directly associated with adverse clinical outcomes in patients with HF. Thus, this study aims to (1) determine the prevalence of persistent hepatic dysfunction at the time of discharge; (2) assess the association between individual LFT panels, as continuous variables, and the patient outcomes; and (3) investigate the relationship between the combination of LFT parameters and long-term clinical outcomes in patients with AHF encompassing various phenotypes.

## 2. Materials and Methods

### 2.1. Study Design

This study analyzed data from the West Tokyo Heart Failure (WET-HF) registry, which consecutively registered patients hospitalized for AHF in 6 tertiary care centers in the Tokyo metropolitan area. The detailed design of the WET-HF registry has been previously reported [6,7]. Briefly, the WET-HF registry is a large, prospective, multicenter cohort registry designed to collect data on the clinical background and outcomes of patients hospitalized for AHF. In this study, we defined AHF as rapid-onset HF or a change in the signs and symptoms of HF requiring urgent therapy and hospitalization based on the Framingham criteria [8]. To ensure the accuracy of the assessment of clinical events, a central study committee that adjudicates the mode of death and the cause of rehospitalization supports the WET-HF registry.

AHF was clinically diagnosed by experienced cardiologists at each institution. Patients presenting with acute coronary syndrome were excluded. The study protocol was approved by the institutional review board at each site, and the research was conducted in accordance with the Declaration of Helsinki. Written or oral informed consent was obtained from each subject before the study.

### 2.2. Patients

We analyzed the data of consecutive AHF patients who were listed in the WET-HF registry between January 2013 and December 2017 as this database did not include data on LFTs before 2013. Of the 2582 patients initially identified, we excluded 112 (4.3%) who died during index hospitalization, 173 who were lost to follow-up (6.7%), and 1139 (44.1%) patients with missing data on LFTs at discharge, leaving 1158 patients for inclusion in the analysis (Figure 1). The results of the comparison of baseline characteristics between the excluded and included patients are shown in Appendix A, with no significant differences found between the two groups except for age (the excluded patients were older), New York Heart Association (NYHA) functional class (higher in the excluded patients), and the usage of diuretics.

### 2.3. Definition of Abnormal Liver Function Tests (LFTs)

Venous blood samples were collected before discharge, and LFTs (i.e., TB, AST or ALT, and ALP) were performed at a central laboratory in each hospital. The cut-off values for each LFT parameter were set as values above the upper limit of the normal range, i.e., at 1.3 mg/dL for TB, 40 IU/L for AST and ALT, and 320 IU/L for ALP. To comprehensively evaluate hepatic function, we focused on the above three parameters and categorized the patients into the following three groups: the normal LFT group (i.e., all parameters were within the normal range), the borderline LFT group (i.e., one of the parameters exceeded the normal range), and the abnormal LFT group (i.e., two or more parameters were above the normal range).

### 2.4. Definitions and Outcomes

Transthoracic echocardiography was performed during the hospitalization after compensation for HF. The cut-off value for the preserved and reduced left ventricular ejection fraction (LVEF) was set at 40%. Ischemic etiology was defined as left ventricular dysfunction with a history of myocardial infarction, a history of coronary revascularization with percutaneous coronary intervention or coronary artery bypass grafting, or at least one major epicardial coronary artery with ≥75% angiographical stenosis on coronary angiography or coronary computed tomography [9,10]. The estimated glomerular filtration rate (eGFR) was determined using the abbreviated Modification of Diet in Renal Disease equation [11]. The primary outcomes were the composite of all-cause mortality and HF readmission after discharge. The secondary outcomes were all-cause mortality and HF readmission, separately.

### 2.5. Statistical Analysis

Data were presented as the mean ± standard deviation or median (interquartile range). Baseline patient characteristics were compared using one-way analysis of variance or the Kruskal–Wallis test for continuous variables and the Pearson chi-square test for categorical variables. Cubic spline analysis was used for evaluating the association between each LFT parameter as a continuous variable and the study’s endpoints.

The log-rank test was used to evaluate the relationship between persistent liver dysfunction at the time of discharge and postdischarge outcomes. A Cox proportional hazard analysis was used to compare the outcomes according to the presence of liver dysfunction, adjusted for the following confounders: sex, age, ischemic etiology, prior hospitalization for HF, atrial fibrillation (AF), diabetes mellitus, chronic obstructive pulmonary disease, LVEF, systolic blood pressure (SBP), hyponatremia (Na < 135 mEq/L), hypoalbuminemia (Alb < 3.0 g/dL), serum levels of blood urea nitrogen (BUN), hemoglobin at discharge, and prescription of renin–angiotensin system inhibitors (i.e., an angiotensin-converting enzyme inhibitor, angiotensin receptor antagonist, or both) and beta-blockers. Subgroup analyses were performed to evaluate the association between abnormal LFTs and each outcome in the prespecified groups: sex, age (cut-off: 80 years), NYHA functional class (I or II and III or IV), ischemic etiology, AF, LVEF (cut-off: 40%), hemoglobin level (cut-off: 10 g/dL), BUN level (cut-off: 30 mg/dL), eGFR (cut-off: 30 mL/min/1.73 m^2^), and B-type natriuretic peptide (BNP) level (cut-off: 300 pg/mL (median)). We analyzed the HF readmission risk using the Fine and Gray model to examine the competing risk of all-cause death. All tests were two-sided, and a *p*-value below 0.05 was considered to indicate statistical significance. Cubic spline analysis was performed using the software R (3.6.1) statistical packages (Foundation for Statistical Computing, Vienna, Austria). Other statistical analyses were performed using IBM SPSS Statistics for Windows, version 25 (IBM Corp.).

## 3. Results

### 3.1. Baseline Characteristics

The mean patient age was 73.9 ± 13.5 years, and 58% of the patients were men. Overall, 121 (10.4%), 190 (16.4%), and 232 (20.0%) patients had persistently elevated TB, AST or ALT, and ALP at the time of discharge, respectively (Appendix A). The baseline characteristics of the patients, according to their LFT results at the time of discharge, are shown in Table 1. Patients with abnormal LFT results were likely to be younger, predominantly men, and have a lower SBP and a higher heart rate. Compared to the normal and borderline LFT groups, the serum sodium levels were significantly lower and BNP levels were higher in the abnormal LFT group. The eGFR level was also higher in the abnormal LFT group. Regarding in-hospital treatment, the usage of vasodilators was significantly lower in the abnormal LFTs group; however, the other medications did not show a significant difference, including infusion diuretics, nitrates, and catecholamines.

### 3.2. Unadjusted Outcomes

There were 196 (16.9%) deaths and 298 (25.7%) HF readmissions during a median follow-up period of 12.4 (interquartile range: 3.6–24.1) months. Figure 2 shows the cubic spline graphs of hazard ratios (HRs) and 95% confidence intervals (CIs) for the associations between each LFT panel and the study endpoint. Although the associations were heterogeneous with regard to HF readmission, there were linear relationships between increased LFT values and the composite outcome and mortality above the upper limit for each LFT panel.

The Kaplan–Meier survival curve shows a higher incidence of the composite outcome in the abnormal LFT group than in the normal LFT group (*p* = 0.008; Figure 3). The abnormal LFT group also demonstrated a higher risk than the borderline LFT group, but the difference was not significant (*p* = 0.149). Furthermore, the abnormal LFT group showed a higher mortality rate than the normal and borderline LFT groups (*p* < 0.001 and 0.003, respectively; Figure 3). There was no significant difference in the rate of HF readmission among the groups.

In sensitivity analyses for the association between each combination of two LFTs parameters and the composite outcome, abnormal values of TB and ALP were significantly associated with higher risk (*p* = 0.012), but not the combination of TB and AST/ALT (*p* = 0.258) and ALP and AST/ALT (*p* = 0.650). For all-cause mortality, on the contrary, abnormal values of TB and AST/ALT (*p* = 0.014) and ALP and AST/ALT (*p* = 0.001) were significantly associated with higher mortality, but not TB and ALP (*p* = 0.055).

### 3.3. Multivariate Adjusted Analysis

In the multivariable Cox proportional hazard model, no single LFT parameter was found to be associated with the outcomes. Furthermore, abnormal LFTs in combination were found to be independently associated with an increased risk of the composite outcome after discharge (adjusted HR, 1.51; 95% CI, 1.08–2.12; *p* = 0.017). A similar result was observed with all-cause mortality (adjusted HR, 2.68; 95% CI, 1.72–4.17; *p* < 0.001) but not with HF readmission (adjusted HR, 0.98; 95% CI, 0.63–1.53; *p* = 0.93). In the prespecified subgroup analyses, a trend towards worse outcomes in the abnormal LFT group was consistently observed (Figure 4) when compared to that in the normal LFT group.

Appendix A shows the analysis of the relationship between the change in LFTs during hospitalization and patient outcomes. By dividing the patients into four groups based on normal or abnormal LFTs at either admission or discharge, the Kaplan–Meier curves indicate that the abnormal LFT group, irrespective of LFTs at admission, appears to have a worse prognosis compared with the normal LFT group at the time of discharge.

## 4. Discussion

In this prospective multicenter study of patients hospitalized for the primary diagnosis of AHF, 37% of the patients continued to have a high level of at least one LFT panel at discharge. Patients with persistently abnormal LFTs (i.e., the elevation of two or more LFT panels above the normal range) showed lower SBP and eGFR and higher BNP levels, which are considered to reflect more severe HF status and partially insufficient decongestion. When assessing any single LFT parameter, the associations with patient outcomes appeared to be heterogeneous, while the combined elevation of LFT parameters was significantly associated with the risk for the postdischarge composite outcome and all-cause mortality but not clearly associated with HF readmission. In the subgroup analysis, the trend towards a worse impact of abnormal LFTs on the composite outcome was consistent in each prespecified patient subgroup. Finally, regardless of the LFT values on admission, abnormal LFTs at the time of discharge appeared to be associated with a high risk of postdischarge outcomes.

LFTs can change markedly in response to treatment during the course of hospitalization. Therefore, when predicting postdischarge outcomes, it is of utmost importance to comprehensively evaluate LFT panels at the time of discharge after in-hospital treatment for hemodynamic derangement [12,13]. However, previous studies have shown that a single LFT panel may not produce consistent results with regard to the prognosis of HF patients [5,14,15,16]. This might be partially related to the heterogeneity of patient backgrounds and also to factors other than hemodynamics [17,18,19]. In the present study, the assessment was carried out by utilizing the combination of the congestion profile (i.e., TB and ALP) and the tissue hypoperfusion profile (i.e., AST/ALT) [20]; the elevation of multiple LFT panels showed a significant relationship with the physical findings of both congestion and tissue hypoperfusion (Appendix A). Furthermore, this was among the strongest predictors of poor outcomes in AHF patients compared to any single LFT panel. Recently, the concept of subclinical (i.e., hemodynamic) congestion and tissue hypoperfusion has received attention and is considered a contributing factor to poor outcomes and, thus, a potential therapeutic target [21]. The comprehensive evaluation of LFTs could provide additional information on the current risk assessment strategies for AHF patients and address the limitations of conventional physical examinations for the risk stratification of patients hospitalized with HF [13]. In the future, further investigation is warranted to establish a way to assess the two pathophysiological manifestations separately to tailor the treatment for HF without invasive direct hemodynamic monitoring systems (i.e., pulmonary artery pressure monitoring).

Our findings are consistent with those of previous studies that have reported a higher prevalence of persistent hepatic dysfunction in patients hospitalized for AHF than in those with chronic HF in the ambulatory setting [2,3,4,5,14,22,23]. This might be explained in part by the insufficient stabilization of the HF status during hospitalization despite the longer length of hospitalization in Japan than that in other countries [10,24,25]. The collaborative report from the Diuretic Optimization Strategy Evaluation in Acute Decompensated Heart Failure (DOSE-AHF) and the Cardiorenal Rescue Study in Acute Decompensated Heart Failure (CARESS-HF) indicated that half of the patients with AHF were discharged with residual clinical congestion [12].

With regard to the association between LFT findings and prognoses, data on persistent hepatic dysfunction during index hospitalization being associated with unfavorable postdischarge outcomes have been limited for AHF patients. Several studies have reported the association between hepatic dysfunction at the time of admission and subsequent short- and middle-term outcomes among these patients. However, these were clinical trials with strict inclusion and exclusion criteria [2,3,4,5]. A secondary analysis of the Acute Study of Clinical Effectiveness of Nesiritide in Decompensated Heart Failure (ASCEND-HF) conducted by Samsky et al. showed that abnormal LFTs at the time of study inclusion (i.e., acute phase) were associated with higher 180-day mortality [5], despite our results showing that the baseline LFTs at the time of admission were not associated with long-term outcomes. Our study is one of the very few studies evaluating LFT data at the time of discharge after in-hospital treatment of AHF patients, including heterogeneous patients with a range of HF phenotypes, such as preserved/reduced EF and low SBPs, in a nonselective setting [15].

Focusing on an individual LFT panel can play an important role in predicting prognosis. Importantly, TB is among the useful indicators for adverse events in AHF patients [2,3,5,14,15,22,23], and it has been included in several risk models for predicting mortality in HF patients. Moreover, TB reflects only the congestion profile, not the effect of tissue hypoperfusion, and, moreover, it does not always reflect hemodynamic instability, as mentioned before. Physiological jaundice (i.e., Gilbert syndrome) is prevalent in 3% to 10% of the general population [17]. Okada et al. previously reported that direct bilirubin, but not TB, is associated with the risk of adverse events in patients hospitalized for AHF [15]. In addition, hepatic dysfunction, with elevated levels of ALP, γ-glutamyl transpeptidase (γ-GTP), and/or AST/ALT, are sometimes caused by hepatocellular injury or cholestasis caused by drugs and intravenous nutrition [18,19,26,27]. In general, medications for AHF patients, including renin–angiotensin system inhibitors and beta-blockers, are often reconditioned during hospitalization. When evaluating the hemodynamics by LFT findings, the possibility of misinterpretation cannot be ruled out when we focus on a single parameter because of a relatively large number of false-positive cases due to the reasons given above (i.e., physiological jaundice or drug-induced liver injury). Similar to previous studies, the present study found the associations between each LFT parameter and the patient outcomes to be heterogeneous and inconsistent. As such, our analysis using a unique definition for abnormal LFTs has the advantage of effectively excluding cases with a single LFT parameter exceeding the upper limit but with unknown clinical importance. Furthermore, to the best of our knowledge, our study can provide additional insight into the clinical relationship between abnormal LFTs and long-term prognosis in AHF patients, unlike previous studies that had a relatively short follow-up period.

Our study has some inherent limitations. First, several patients were excluded because they lacked LFT data, although there were no differences in patient characteristics between the included and excluded patients except for some parameters (i.e., age and NYHA functional class). Second, we did not have data on factors associated with hepatic impairment, such as primary liver diseases or drug abuse. Nonetheless, the prevalence of extremely high TB (>3 mg/dL) at discharge, which can indicate the presence of primary liver diseases, was quite low in our study population (8 per 1158 patients). Third, we could not find any significant relationship between abnormal LFTs and HF readmission. Previous studies predicting clinical outcomes of HF patients showed the disparity between predictors of mortality and HF readmission among these patients [28,29]. Predictors of HF with acceptable discriminative capability remain challenging, as there are various factors that can influence HF hospitalization, including disease severity, comorbidity, nutritional and physical status (frailty), treatment adherence, and socioeconomic status. These complex risk factors for HF readmission might explain the two unrelated events observed in our analysis. Fourth, γ-GTP, one of the important enzymes in a congestion profile, was not assessed because it was not included in the present registry. Finally, because of the nature of the study design, there is a possibility that unknown confounders influenced our results.

## 5. Conclusions

In this contemporary Japanese AHF cohort, approximately 8.6% of patients had persistently high LFT parameters (i.e., a combination of congestion and tissue hypoperfusion profiles) at the time of discharge. Among AHF patients, persistent hepatic dysfunction, even after in-hospital treatment, was significantly associated with long-term adverse outcomes. Our method of evaluating a combination of LFT panels can be helpful in the risk stratification of HF patients and the tailoring of postdischarge treatment in the HF population.

## Figures and Tables

**Figure 1 jcm-10-01730-f001:**
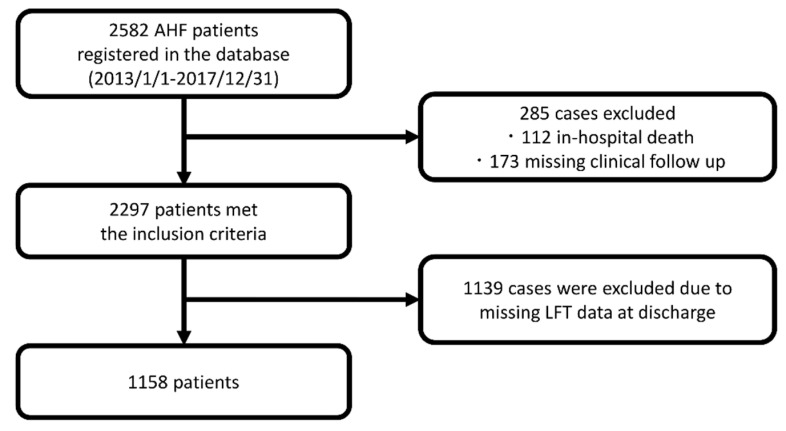
Study patient flow chart. Abbreviations: AHF = acute heart failure; LFT = liver function test.

**Figure 2 jcm-10-01730-f002:**
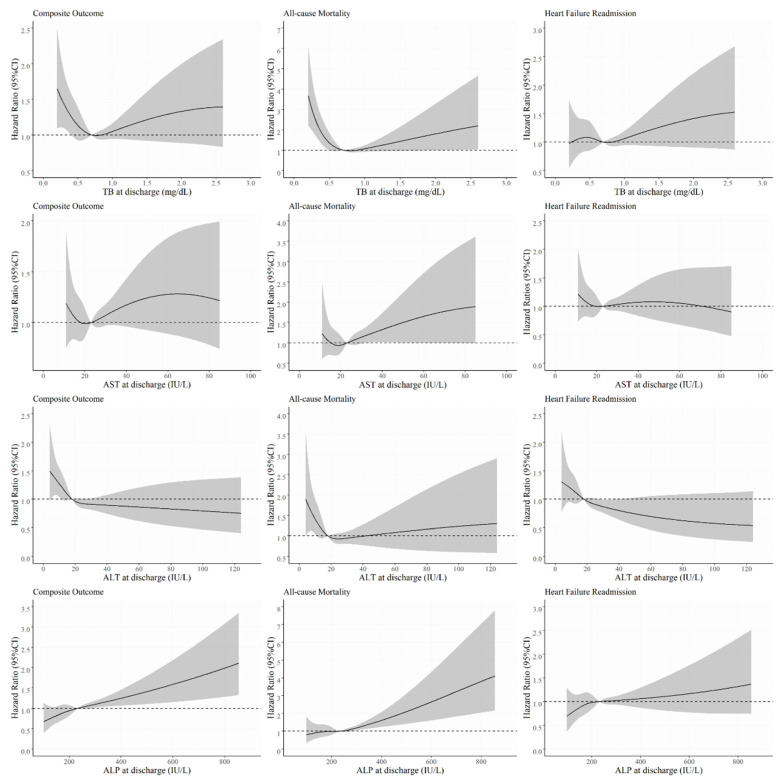
Cubic spline graphs for the associations between each LFT panel and patient outcomes (black curve for HR and gray area for 95% CI). Abbreviations: HR = hazard ratio; CI = confidence interval; TB = total bilirubin; AST = aspartate aminotransferase; ALT = alanine aminotransferase; ALP = alkaline phosphatase.

**Figure 3 jcm-10-01730-f003:**
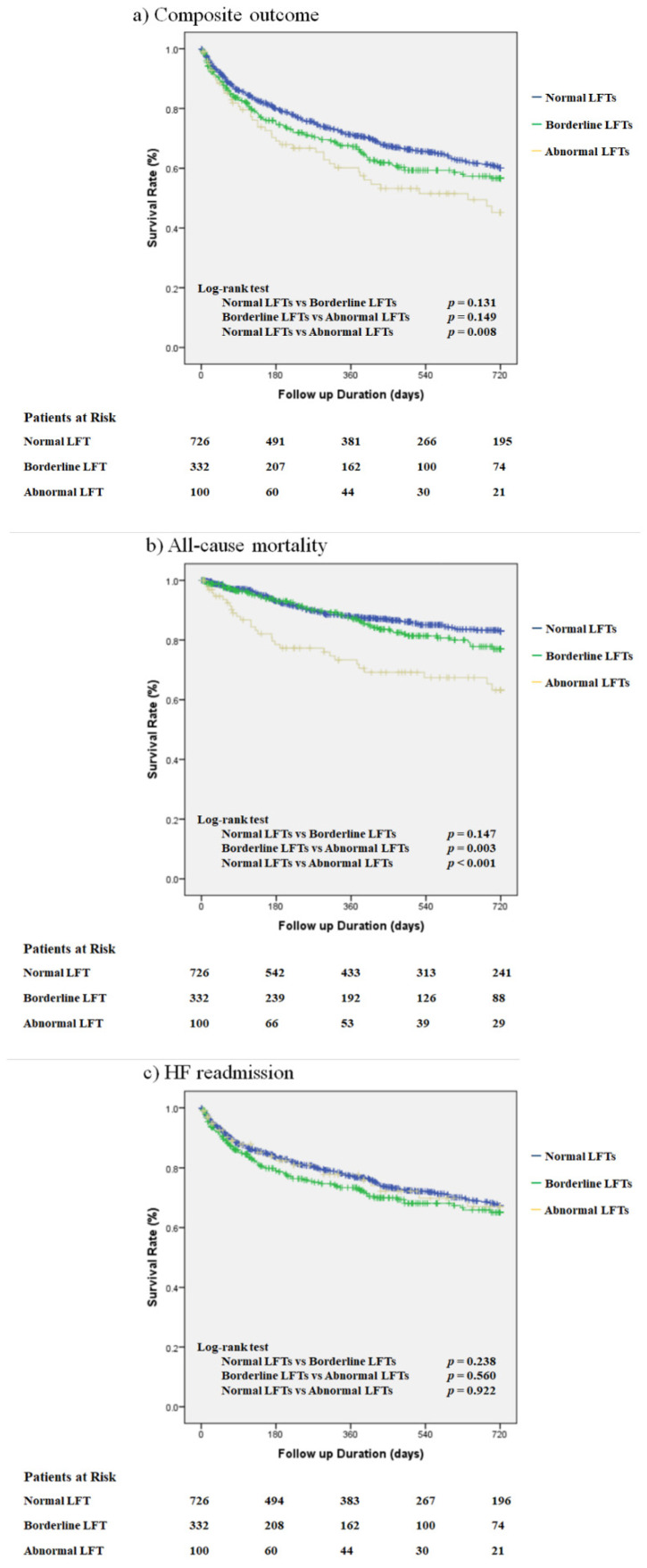
Kaplan–Meier survival curve for each outcome. (**a**) Composite outcome; (**b**) all-cause mortality; (**c**) HF readmission. Abbreviations: HF = heart failure; LFT = liver function test.

**Figure 4 jcm-10-01730-f004:**
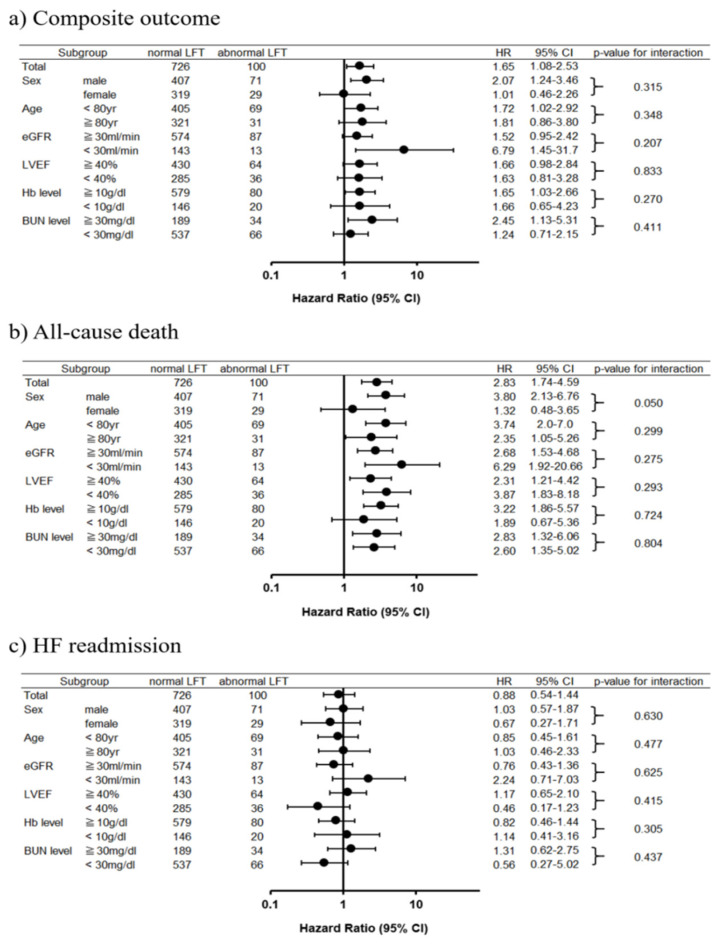
Forest plots of each outcome. (**a**) Composite outcome; (**b**) all-cause mortality; (**c**) HF readmission. Abbreviations: HF = heart failure; LFT = liver function test; eGFR = estimated glomerular filtration rate; LVEF = left ventricular ejection fraction; BUN = blood urea nitrogen.

**Table 1 jcm-10-01730-t001:** Baseline characteristics.

Variables	Normal LFTs *n* = 726	Borderline LFTs *n* = 332	Abnormal LFTs *n* = 100	*p*-Value
Age, years	78 (67–84)	76 (65–82)	72 (62–80)	0.003
Male, %	56.1	59.6	71.0	0.015
BMI, kg/m^2^	22.9 (20.4–26.0)	23.1 (20.6–26.0)	23.1 (20.7–26.0)	0.56
SBP at discharge, mmHg	112 (100–125)	106 (98–120)	107 (98–120)	<0.001
HR at discharge, bpm	70 (62–79)	72 (64–80)	74 (66–84)	0.014
NYHA class at discharge				0.55
Class I, %	29.3	29.6	24.0	
Class II, %	58.3	58.3	58.0	
Class III, %	11.3	10.3	17.0	
Class IV, %	1.1	1.8	1.0	
LVEF, %	45.0 (32.0–59.0)	45.0 (31.0–56.0)	47.0 (30.0–58.0)	0.61
Ischemic etiology, %	31.5	24.1	20.0	0.007
Comorbidities				
Prior admissions for HF, %	25.5	25.9	30.0	0.63
Hypertension, %	66.4	65.1	61.0	0.55
Hyperlipidemia, %	39.9	39.6	35.0	0.64
Diabetes, %	32.6	31.9	28.0	0.65
Atrial fibrillation, %	44.8	53.0	51.0	0.035
Stroke, %	14.3	17.2	16.0	0.48
COPD, %	5.1	2.1	7.0	0.037
Hemodialysis, %	2.6	1.2	1.0	0.24
Laboratory findings at admission				
Hemoglobin, mg/dL	11.8 (10.3–13.6)	12.5 (10.8–14.1)	12.6 (10.4–14.1)	0.005
Sodium, mEq/L	140 (137–142)	140 (137–142)	139 (135–141)	0.016
Cr, mg/dL	1.4 ± 1.5	1.2 ± 0.7	1.4 ± 1.1	0.035
BUN, mg/dL	21.2 (16.6–30.1)	22.1 (16.1–31.5)	23.8 (17.3–33.7)	0.042
eGFR, mL/min/1.73 m^2^	49.8 ± 24.0	50.9 ± 20.2	51.5 ± 23.3	0.65
TB, mg/dL	0.8 (0.6–1.1)	1.0 (0.7–1.5)	1.5 (0.8–2.0)	<0.001
AST, IU/L	31.0 (23.0–46.5)	36.0 (26.0–55.0)	42.0 (30.0–74.0)	0.64
ALT, IU/L	21.0 (13.0–37.0)	27.0 (17.0–47.0)	32.0 (19.3–67.3)	0.23
ALP, IU/L	245 (196–298)	297 (207–412)	389 (273–528)	<0.001
Albumin, g/dL	3.6 (3.3–3.9)	3.6 (3.3–3.9)	3.5 (3.3–3.8)	0.13
BNP, pg/mL	751 (419–1370)	573 (365–1010)	992 (489–1880)	0.001
Laboratory findings at discharge				
Hemoglobin, mg/dL	11.7 (10.3–13.1)	12.4 (10.6–14.0)	12.3 (10.3–14.0)	<0.001
Sodium, mEq/L	139 (137–141)	139 (137–141)	138 (135–140)	<0.001
Cr, mg/dL	1.4 ± 1.5	1.2 ± 1.0	1.2 ± 0.9	0.49
BUN, mg/dL	21.5 (16.0–30.3)	22.3 (16.4–30.8)	25.1 (17.4–33.5)	0.051
eGFR, mL/min/1.73 m^2^	49.7 ± 22.9	52.7 ± 22.1	55.5 ± 27.1	0.027
TB, mg/dL	0.6 (0.5–0.9)	0.8 (0.6–1.2)	1.2 (0.7–1.7)	<0.001
AST, IU/L	21.0 (17.0–26.0)	27.0 (21.0–39.0)	44.0 (31.3–55.0)	<0.001
ALT, IU/L	15.0 (11.0–22.0)	23.0 (15.0–41.0)	42.5 (25.0–67.5)	<0.001
ALP, IU/L	214 (175–253)	287 (205–382)	407 (336–543)	<0.001
Albumin, g/dL	3.5 (3.2–3.8)	3.6 (3.2–3.9)	3.4 (3.1–3.8)	0.18
BNP, pg/mL	271 (137–522)	239 (104–447)	389 (247–663)	0.023
In-hospital treatment				
Diuretic infusion, %	69.3	73.5	68.0	0.33
Vasodilator, %	60.9	57.8	47.0	0.028
Catecholamine, %	16.4	17.5	23.0	0.27
NPPV, %	24.7	18.4	20.0	0.061
Intubation, %	3.4	3.6	6.0	0.45
Prescription at discharge				
Diuretics, %	73.3	77.1	78.0	0.31
RAS inhibitor, %	59.6	59.6	45.0	0.018
MRA, %	34.2	36.7	39.0	0.52
Beta blocker, %	77.4	80.4	71.0	0.13
OAC, %	53.2	62.3	71.0	0.001

Values are mean ± SD or median (interquartile range). Abbreviations: LFT = liver function test; BMI = body mass index; SBP = systolic blood pressure; HR = heart rate; NYHA = New York Heart Association; LVEF = left ventricular ejection fraction; HF = heart failure; COPD = chronic obstructive pulmonary disease; BUN = blood urea nitrogen; eGFR = estimated glomerular filtration rate; TB = total bilirubin; AST = aspartate aminotransferase; ALT = alanine aminotransferase; ALP = alkaline phosphatase; BNP = B-type natriuretic peptide; NPPV = noninvasive positive pressure ventilation; RAS = renin–angiotensin system; MRA = mineralocorticoid receptor antagonist; OAC = oral anticoagulant.

## Data Availability

The data presented in this study are available on request from the corresponding author.

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
