# Peer review of "Abnormal Liver Function Tests and Long-Term Outcomes in Patients Discharged after Acute Heart Failure"

_jcm, 2021, doi:10.3390/jcm10081730_

Round 1
Reviewer 1 Report
On Definitions and outcomes:
Why was the value of 75% stenosis determined as patient with ischaemic cardiomyopathy , it was deteted by CT or Cath? Was and ischaemia test performed?
Table 1 :revise: NYHA class is etter NYHA I , II, III and IV (Dolgin M, Association NYH, Fox AC, Gorlin R, Levin RI, New York Heart Association. Criteria Committee. Nomenclature and criteria for diagnosis of diseases of the heart and great vessels. 9th ed. Boston, MA: Lippincott Williams and Wilkins; March 1, 1994.)
Author Response
Response to Reviewer 1
#1. On Definitions and outcomes: Why was the value of 75% stenosis determined as patient with ischaemic cardiomyopathy, it was detected by CT or Cath? Was and ischaemia test performed?
Response:
- We truly appreciate the time and effort taken by the reviewers to review our manuscript. The cutoff of 75% stenosis was dirven from previous studies that defined ischemic cardiomyopathy as left ventricular dysfunction (left ventricular ejection fraction ≤40%) with a history of myocardial infarction, percutaneous coronary intervention, coronary artery bypass grafting, or at least 1 major epicardial coronary artery with ≥75% stenosis (Bart BA, et al. Clinical determinants of mortality in patients with angiographically diagnosed ischemic or nonischemic cardiomyopathy. J Am Coll Cardiol 1997;30(4):1002-8). This defenition has been applied to other HF-related multicenter registries in Japan (Shiraishi Y, et al. J Am Heart Assoc 2018;7(18):e008687). For clarification,we have added the above information in the revised ‘Methods’ section.
Changes: The description of definition for ischemic cardiomyopathy was added (Line 94-98):
“Ischemic etiology was defined as left ventricular dysfunction with a history of myocardial infarction, history of coronary revascularization with percutaneous coronary intervention or coronary artery bypass grafting, or at least 1 major epicardial coronary artery with ≥75% angiographical stenosis on coronary angiography or coronary computed tomography”
#2. Table 1: revise: NYHA class is lettered I, II, III and IV (Dolgin M, Association NYH, Fox AC, Gorlin R, Levin RI, New York Heart Association. Criteria Committee. Nomenclature and criteria for diagnosis of diseases of the heart and great vessels. 9th ed. Boston, MA: Lippincott Williams and Wilkins; March 1, 1994.)
Response:
- Thank you for pointing this out. We have changed lettering of NYHA classification (1, 2, 3, and 4 to class I, II, III, and IV).
Reviewer 2 Report
Here, Miyama and colleagues examined 1,158 patients with acute heart failure (AHF) from a prospective multicenter registry in Japan. From these subjects that were successfully discharged (mean age, 73.9±13.5 years; men, 58%), the Authors parameters monitoring liver function, such as total bilirubin, aspartate aminotransferase or alanine aminotransferase, and alkaline phosphatase.
The main conclusion drawn by the authors is that in AHF patients, the elevation of two or more of these liver function parameters is associated with long-term adverse outcomes.
No correlation was observed with the readmission to the hospital; still, the study provides some interesting insights on the relevance of persisting liver dysfunction on the management of patients with AHF.
I have a few comments to make.
- Rationale of the study (Introduction). The Authors stated that “previous studies have focused largely on a single abnormality of LFTs, such as total bilirubin (TB), which may be elevated due to factors other than hemodynamic instability (e.g., drug-induced liver injury, physiological jaundice, and malnutrition). What evidence do the authors have that the chosen parameters, i.e., aspartate aminotransferase [AST] or alanine aminotransferase [ALT], and alkaline phosphatase [ALP] do not suffer from the same limitations?
- The higher prevalence of persistent hepatic dysfunction in patients hospitalized for AHF with respect to those with chronic HF in the ambulatory setting appears to be a recurrent, significant observation. The authors speculate that “this might be explained in part by the insufficient stabilization of the HF status during hospitalization despite the longer length of hospitalization in Japan than that in other countries”. Could the authors provide more info on the therapeutic measures taken to reduce liver congestion during hospitalization? Is this based only on the use of diuretics?
- The Discussion does not attempt to explain why patients with “abnormal LFT results were likely to be younger, predominantly men, and have a lower SBP and a higher heart rate.” It would be nice if the authors can spend more paper on these relevant findings, particularly in view of the reported data that “compared to the normal and borderline LFT groups, the serum sodium levels were significantly lower, and BNP levels were higher in the abnormal LFT group.”
- Same for the eGFR level that was also higher in the abnormal LFT group. How do the authors explain this phenomenon?
Author Response
Response to Reviewer 2
Here, Miyama and colleagues examined 1,158 patients with acute heart failure (AHF) from a prospective multicenter registry in Japan. From these subjects that were successfully discharged (mean age, 73.9±13.5 years; men, 58%), the Authors parameters monitoring liver function, such as total bilirubin, aspartate aminotransferase or alanine aminotransferase, and alkaline phosphatase.
The main conclusion drawn by the authors is that in AHF patients, the elevation of two or more of these liver function parameters is associated with long-term adverse outcomes.
No correlation was observed with the readmission to the hospital; still, the study provides some interesting insights on the relevance of persisting liver dysfunction on the management of patients with AHF.
I have a few comments to make.
#1. Rationale of the study (Introduction). The Authors stated that “previous studies have focused largely on a single abnormality of LFTs, such as total bilirubin (TB), which may be elevated due to factors other than hemodynamic instability (e.g., drug-induced liver injury, physiological jaundice, and malnutrition). What evidence do the authors have that the chosen parameters, i.e., aspartate aminotransferase [AST] or alanine aminotransferase [ALT], and alkaline phosphatase [ALP] do not suffer from the same limitations?
Response:
- First and foremost, thank you for your time and effort in providing comments for our manuscript. We had based the selection of LFTs on the availability of the tests across the institutions (which leads to high generalizability of the study result). Moreover, multiple previous studies have assessed the liver function based on these parameters (Samsky MD, et al. Eur J Heart Fail. 2016;18(4):424-32). However, as pointed out, elevation of LFTs do not universally reflect hemodynamic instability and cannot rule out the influence of other factors (hence, it was difficult to define the relationship between adverse outcomes of HF patients and abnormal liver function based on a single LFT parameter). These parameters can be elevated drug-induced hepatocellular injury or cholestasis (Hepatology 2002;36(2):451-5, Hepatology 2011;53(4):1377-87, Clin Liver Dis 2016;2(1):159-76). Moreover, AST and ALP is known to present not only in liver but also in other organs including cardiac muscle, skeletal muscle, kidney, and bone. As such the heterogenous pathology and low tissue specificity can lead to the same limitation discussed in TB. Based on the above consideration, we added the above description to the revised ‘Introduction’ and ‘Discussion’ sections.
- However, despite the above limitation, we believe that our main hypothesis that the multi-parameter evaluation of LFT enabled better assessment of the overall condition of the HF is valid and supported by our primary analysis that demonstrated higher risk of adverse outcomes in patients with ≥2 LFT parameter values above the normal range.
Changes: The introduction section now contains the following sentences:
“However, previous studies have focused largely on a single abnormality of LFTs, such as total bilirubin (TB), aspartate aminotransferase (AST) or alanine aminotransferase (ALT), and alkaline phosphatase (ALP) which may be elevated due to factors other than hemodynamic instability (e.g., drug-induced liver injury, physiological jaundice, and malnutrition).” (Line 36-40)
In addition, the discussion section now contains the following sentences:
“In addition, hepatic dysfunction with elevated levels of ALP, γ-glutamyl transpeptidase (γ-GTP), and/or AST/ALT are sometimes caused by hepatocellular injury or cholestasis caused by drugs and intravenous nutrition.” (Line 268-270)
#2. The higher prevalence of persistent hepatic dysfunction in patients hospitalized for AHF with respect to those with chronic HF in the ambulatory setting appears to be a recurrent, significant observation. The authors speculate that “this might be explained in part by the insufficient stabilization of the HF status during hospitalization despite the longer length of hospitalization in Japan than that in other countries”. Could the authors provide more info on the therapeutic measures taken to reduce liver congestion during hospitalization? Is this based only on the use of diuretics?
The Discussion does not attempt to explain why patients with “abnormal LFT results were likely to be younger, predominantly men, and have a lower SBP and a higher heart rate.” It would be nice if the authors can spend more paper on these relevant findings, particularly in view of the reported data that “compared to the normal and borderline LFT groups, the serum sodium levels were significantly lower, and BNP levels were higher in the abnormal LFT group.” Same for the eGFR level that was also higher in the abnormal LFT group. How do the authors explain this phenomenon?
Response:
- Thank you for your insightful comments and suggestions. The table below is a summary of the in-hospital treatment for each LFT group. There were no apparent differences in in-hospital treatment patterns between the three groups (normal vs. borderline vs. abnormal LFT groups), other than the usage of intravenous vasodilators (less frequently used in abnormal LFT group, p=0.028).
Table. In-hospital treatment patterns by the results of baseline liver function test.
|
Variables |
Normal LFTs N=726 |
Borderline LFTs N=332 |
Abnormal LFTs N=100 |
P value |
|
Loop diuretic, IV, % |
69.3 |
73.5 |
68.0 |
0.33 |
|
Vasodilator, IV, % |
60.9 |
57.8 |
47.0 |
0.028 |
|
Catecholamine, IV, % |
16.4 |
17.5 |
23.0 |
0.27 |
|
NPPV, % |
24.7 |
18.4 |
20.0 |
0.061 |
|
Intubation, % |
3.4 |
3.6 |
6.0 |
0.45 |
- Although it is difficult to fully identify the association of the usage of intravenous vasodilator with LFT, we speculate that the abnormal LFT group showed a lower systolic blood pressure and such patients were likely to have a greater severity, possibly with a difficulty of achieving decongestion. This is consistent with our findings on the relationship with physical findings and congestion / tissue hypoperfusion profiles (Table S2 and S3).
- As for the co-assessment of the renal function, better renal function at the time of discharge were seen in the abnormal LFT group. To address the reviewer’s comments, we performed an additional analysis on changes in eGFR and BNP levels from admission to discharge and their associations with LFTs. We defined improved renal function (IRF) as 20% or more improvement in eGFR relative to admission and adequate decongestion as a 40% or more decrease in BNP relative to admission based on a previous study (Wettersten N, et al. Eur J Heart Fail 2021 [Online ahead or print]).
In this subanalysis, IRF was frequently observed in the abnormal LFT group, while adequate decongestion (BNP improvement) was less frequently seen (IRF; 30.0% in the abnormal LFTs vs. 20.4% in the normal LFTs, P = 0.028, BNP improvement; 61.9% in the abnormal LFTs vs. 69.1% in the normal LFTs, P = 0.35, respectively). The trajectory of decreasing BNP which indicates effective decongestion during the AHF treatment, and may have a better predictive value for post-discharge survival. In addition, lower sodium levels and systolic blood pressures in the abnormal LFT group are considered to reflect severe HF status and potentially residual congestion.
- Accordingly, we have added the following statements to the revised manuscript.
Changes: The discussion section now contains the following sentence: “Patients with persistently abnormal LFTs (i.e., the elevation of 2 or more LFT panels above the normal range) showed a lower SBP and eGFR, and a higher BNP level, which are considered to reflect more severe HF status and partially insufficient decongestion.” (Line 205-208)